# Species-specific modulation of food-search behavior by respiration and chemosensation in Drosophila larvae

Daeyeon Kim[1,2], Mar Alvarez[3], Laura M Lechuga[3], Matthieu Louis[1,2,4,5]*

[1]EMBL-CRG Systems Biology Research Unit, Centre for Genomic Regulation, The Barcelona Institute of Science and Technology, Barcelona, Spain; [2]Universitat Pompeu Fabra, Barcelona, Spain; [3]Nanobiosensors and Bioanalytical Applications Group, Catalan Institute of Nanoscience and Nanotechnology, CSIC and The Barcelona Institute of Science and Technology, CIBER-BBN, Barcelona, Spain; [4]Neuroscience Research Institute, University of California, Santa Barbara, Santa Barbara, United States; [5]Department of Molecular, Cellular, and Developmental Biology, University of California, Santa Barbara, Santa Barbara, United States

**Abstract** Animals explore their environment to encounter suitable food resources. Despite its vital importance, this behavior puts individuals at risk by consuming limited internal energy during locomotion. We have developed a novel assay to investigate how food-search behavior is organized in *Drosophila melanogaster* larvae dwelling in hydrogels mimicking their natural habitat. We define three main behavioral modes: resting at the gel's surface, digging while feeding near the surface, and apneic dives. In unstimulated conditions, larvae spend most of their time digging. By contrast, deep and long exploratory dives are promoted by olfactory stimulations. Hypoxia and chemical repellents impair diving. We report remarkable differences in the *dig-and-dive* behavior of *D. melanogaster* and the fruit-pest *D. suzukii*. The present paradigm offers an opportunity to study how sensory and physiological cues are integrated to balance the limitations of dwelling in imperfect environmental conditions and the risks associated with searching for potentially more favorable conditions.

DOI: https://doi.org/10.7554/eLife.27057.001

*For correspondence:
mlouis@lifesci.ucsb.edu

**Competing interests:** The authors declare that no competing interests exist.

## Introduction

The natural habitats populated by individual species of the Drosophila group comprise a wide range of food resources ranging from fermenting fruits to vegetables (*Hansson and Stensmyr, 2011*). The common lab species, *Drosophila melanogaster*, is a cosmopolitan species that breeds on multiple sites, including tomatoes (*Mccoy, 1962*; *Jaenike, 1983*). Most of the substrates where larvae grow are characterized by soft semiliquid structures similar to hydrogels. When placed on a slice of fresh tomato, *D. melanogaster* larvae readily locate the soft middle layer of the fruit, called the locular gel, and dig into it (*Figure 1A and A'*). The marked preference for the locular gel over the more external mescocarp layer appears to be mostly due to the soft composition of the locular gel rather than potential differences in nutritive values (*Figure 1—figure supplement 1*). Digging behavior is commonly observed in regular food vials and in behavioral assays (*Figure 1B and C*). Not surprisingly, digging takes place when the softness of low-density-agarose gels is similar to the natural habitats of *Drosophila melanogaster* (for stiffness measurements related to tomato pulp, see *Grant et al., 2012*; *Li et al., 2012*). Hereafter, exploration of the substrate along the vertical axis will be referred to as 'dives'. As nutritional and pH conditions vary along the depth of a fruit, exploratory dives permit larvae to search for conditions that satisfy their metabolic and physiological needs.

Diving is also thought to facilitate dispersal through the environment while minimizing predator encounters at the surface (*Godoy-Herrera, 1977*, *1994*).

Digging-and-diving behavior fulfills additional functions. Cooperative digging enables populations of larvae to liquefy food sources through the effects of pre-digestive enzymes secreted in the saliva (*Gregg et al., 1990*; *Chandrashekar et al., 2009*). Like their adult counterparts (*Enjin et al., 2016*), *Drosophila* larvae are highly sensitive to humidity (*Benz, 1956*). Dwelling into semiliquid media prevents desiccation. It also enables larvae to hide from strong daylight and from predators (*Hwang et al., 2007*; *Xiang et al., 2010*; *Robertson et al., 2013*). In contrast with detailed analysis of foraging and odor-driven responses elicited on flat surfaces of agarose (*Sokolowski, 1985*; *Cobb, 1999*; *Kreher et al., 2008*; *Louis et al., 2008*; *Gershow et al., 2012*), the organization of digging and diving behaviors into 3D substrates remains poorly understood. Here, we present a new experimental system, called *dig-and-dive* assay, to study food-search behavior in naturalistic conditions. We built a transparent chamber filled with an agarose gel in which the digging and diving of single larvae can be monitored (*Figure 2A*). During their exploration of the chamber, larvae dive into the hydrogel substrate for several minutes at depths larger than three times their body length.

Since larvae are air-breathing animals that cannot oxygenate underwater (*Manning and Krasnow, 1993*), diving in hydrogels poses a physiological challenge. Individuals that failed to return to the surface after a few minutes stopped moving and the majority of them drowned. In the present work, we examined how the initiation of apneic dives is conditioned by physiological and sensory signals. First, we studied the effect of hypoxia on diving behavior. By partially blocking the respiratory tracks of larvae, we found evidence that the level of oxygenation determines the propensity of larvae to dive. This result suggests that diving is regulated by a cost-benefit balance, which combines internal physiological signals (e.g., oxygenation level and possibly hunger) with external cues (e.g., presence of food odor). Second, we studied the effect of the hardness of the substrate on the control of diving behavior. Third, we asked whether the detection of an attractive odor is sufficient to modulate food-search behavior and, more specifically, to promote diving. Our results demonstrate that larvae perceive and respond to liquid-borne odorant molecules. In line with the idea that diving relies on a cost-benefit balance adjusted by sensory signals, we found that the duration of apneic dives increases upon detection of an attractive odor. We compared the dig-and-dive behavior of *D. melanogaster* with a second species of the Drosophila group, the fruit pest *D. suzukii* (*Lee et al., 2011*). Consistent with the innate preference of *D. suzukii* for fresh fruits, larvae of this species are capable of digging into harder substrates than *D. melanogaster*. Our results indicate that *D. suzukii* is also more resistant to hypoxia. Finally, we report the existence of species-specific differences in the modulatory effects of attractive and repulsive odors on the dig-and-dive behavior of *D. melanogaster* and *D. suzukii*.

## Results

### Larvae forage in hydrogels by surfacing, digging, and diving

To study food-search behavior in hydrogel structures that resemble the natural habitats of larvae, we created a miniature device using a micro-milling method and polydimethylsiloxane (PDMS) elastomer to build a transparent chamber (*Figure 2A*). The whole assay consists of three sub-regions: a closeable channel at the top to load a larva, a wide air chamber, and a long and narrow agarose chamber with a length of 12 mm that corresponds to approximately three-body lengths of larvae at the third developmental instar (Materials and methods). The width of the agarose chamber is 3 mm to permit larvae to turn around in the channel and to resurface. The dig-and-dive assay was designed to limit the exploration of the chamber along the horizontal axis (*Figure 2—figure supplement 1*). Larvae were constrained to the chamber by a silicon cap inserted through the top of the loading channel. Although the chamber was closed, oxygen was supplied through two thin air channels on the side of the cap (*Figure 2B*). In this transparent assay, the behavior of individual larvae was recorded with a CCD camera. A computer vision algorithm was then applied to extract the position of the centroid of the larva during the course of a 15 min experiment.

Larvae displayed a behavior that strongly depended on their position in the assay chamber. At the surface, larvae tended to be inactive. While immersed in the agarose gel, larvae demonstrated two different types of behavior: right under the surface, they oriented head down and kept their

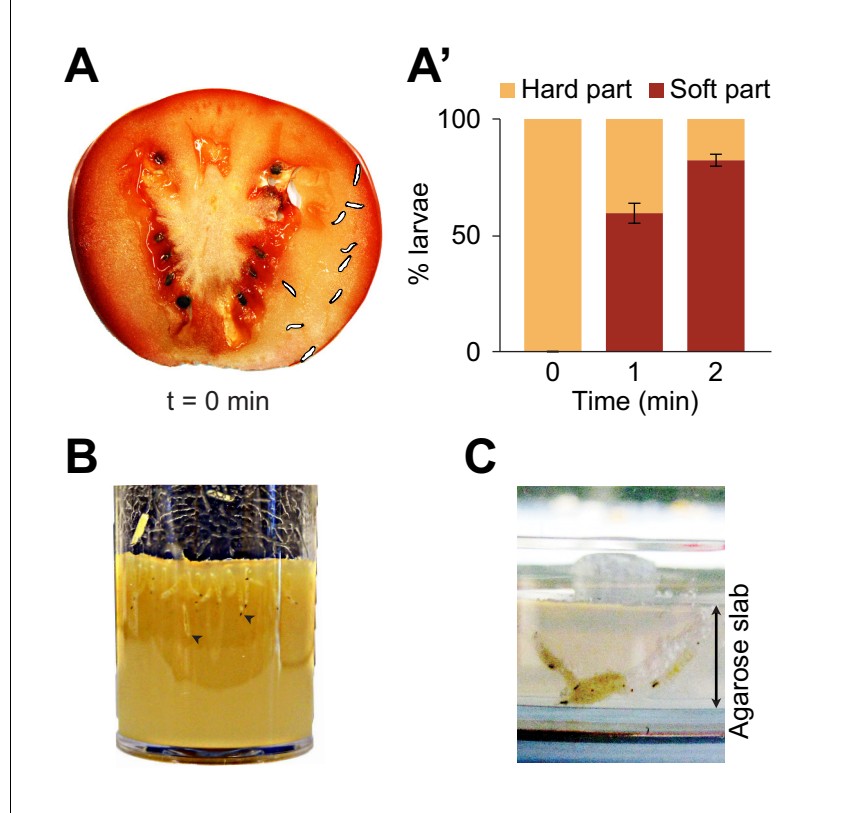

**Figure 1.** Foraging behaviors of larvae on a fruit and in controlled laboratory conditions. Digging and diving behaviors of larvae in (**A**) a tomato slice, (**B**) a vial of lab food (arrowheads indicate diving larvae), and (**C**) a plain agarose slab (2%, w/v). Larvae introduced on hard agarose slabs as shown in (**C**) initiated burrowing after having foraged for more than 30 min on the surface. In (**A**), the outline of the individual larvae was drawn by hand upon careful inspection of pictures to increase the contrast with the background. In (**A'**), the percentage of larvae located in the hard part (mesocarp) and soft part (locular gel) of the tomato slice were quantified over time upon introduction of larvae ($n$ = 9 or 10 individuals) on the hard part of the fruit (bars report means ± s.e.m., $n$ = 8 trials). One-sample Student's $t$-test of the mean fraction of larvae found in the soft part of the tomato at time point t = 2 min compared to 50%: p<0.001. For more information about the statistics, see **Supplementary file 1**.
DOI: https://doi.org/10.7554/eLife.27057.002

The following figure supplement is available for figure 1:

**Figure supplement 1.** Spatial preferences elicited by substrates of different hardnesses and nutritional values.
DOI: https://doi.org/10.7554/eLife.27057.003

spiracles in contact with the air at the gel's interface (**Figure 2B**). We term this 'snorkeling' behavior *digging*. Larval digging is commonly observed in transparent food vials in the lab (**Figure 1B**) (**Sokolowski, 1982**). Digging permits larvae to feed and breathe while staying hidden from predators at the surface. When the posterior spiracles of the larva went below the surface of the gel, larvae initiated apneic dives along the vertical axis of the chamber (**Figure 2C**). Individual dives lasted up to a couple of hundred seconds during which the larva remained active until it returned to the surface. To classify the main behavioral modes observed in the dig-and-dive assay, we computed the probability density function (PDF) of the vertical position of the centroid of the larva (variable denoted as Z) in the gel (**Figure 2B**, left panel). The mean of the PDF was located 2.7 mm below the surface of the gel. The PDF displayed a long tail toward each end of the chamber. Accordingly, we divided the PDF into three different regions based on two thresholds, $Z_{digging} = 0$ and $Z_{diving} = -3.6$ mm (see Materials and methods). In the upper tail of the PDF ($Z > Z_{digging}$), larvae moved in the air chamber above the surface of the gel. We defined this behavioral mode as *surfacing*. In the central region of the PDF ($Z_{diving} < Z \leq Z_{digging}$), larvae were engaged in digging with at least half of their bodies embedded in the gel. As larvae appeared to be actively feeding through the contraction of their

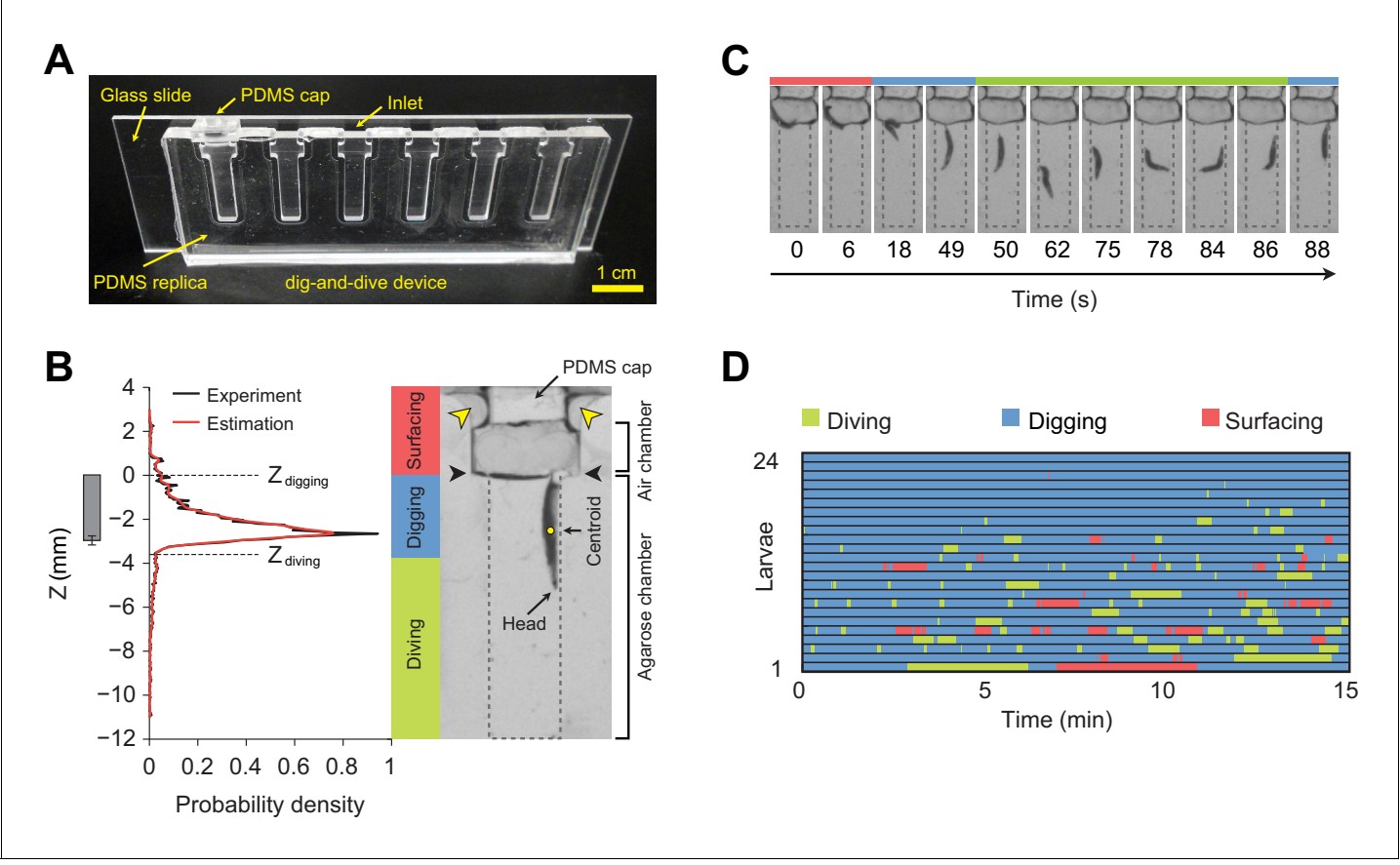

**Figure 2.** Dig-and-dive assay and definition of behavioral modes. (A) Picture of the dig-and-dive device. The dimensions of the assay are given in *Figure 2—figure supplement 1*. (B) Probability density function (PDF) of larval centroid depths (Z) observed in 0.4% plain agarose gel. Kernel density estimation was applied to smoothen the PDF (red). The thresholds on the depth for classifying digging and diving behavioral modes are $Z_{digging}$ = 0, $Z_{diving}$ = −3.6 mm, respectively. The left bar represents the distribution over larval centroid position plus the length of the posterior spiracles (mean ± s. d., $n$ = 24 pictures). Typical posture of a larva engaged in digging behavior. Yellow arrowheads indicate two lateral air channels on the side of the removable cap (200 µm in thickness). Black arrowheads represent the interface between the agarose gel and the air. (C) A sequence of larval postures corresponding to surfacing (red) followed by digging (blue), diving (green) and a new digging episode (blue). After a dive, larvae do not necessarily resurface. (D) Ethogram over time of 24 larvae. Each row of the ethogram corresponds to a different animal. The time course of the behavioral state of a larva is represented according to the color code at the top of the panel. Trials were sorted by increasing total dive times. The average number of dives per trial is 4.1 ± 3.2 (mean ± s.d.). More information about the statistics is given in *Supplementary file 1*.

DOI: https://doi.org/10.7554/eLife.27057.004

The following figure supplement is available for figure 2:

**Figure supplement 1.** Schematic diagram of the assay device.

DOI: https://doi.org/10.7554/eLife.27057.005

mouth hook (*Sokolowski, 1982*; *Green et al., 1983*; *Schoofs et al., 2010*; *Wang et al., 2013*), digging can be associated with the exploitation of the medium (*Cohen et al., 2007*). The bottom part of the PDF (Z ≤ $Z_{diving}$) corresponded to *diving* behavior during which oxygenation at the surface was interrupted and larvae became hypoxic (*Morton, 2011*). Diving can be associated with the exploration of the medium. Behaviors of individual larvae tested in the dig-and-dive assay were annotated based on these three elementary modes (*Figure 2D*).

## Substrate hardness affects exploratory behavior

Building on our observation that larvae display a strong preference for the softer internal layer of tomato slices (*Figure 1A and A'*, *Figure 1—figure supplement 1*), we investigated the effects of the stiffness of the substrate on diving behavior (*Figure 3*). We took advantage of the fact that the stiffness of an agarose gel is proportional to the percentage of agarose mixed with water

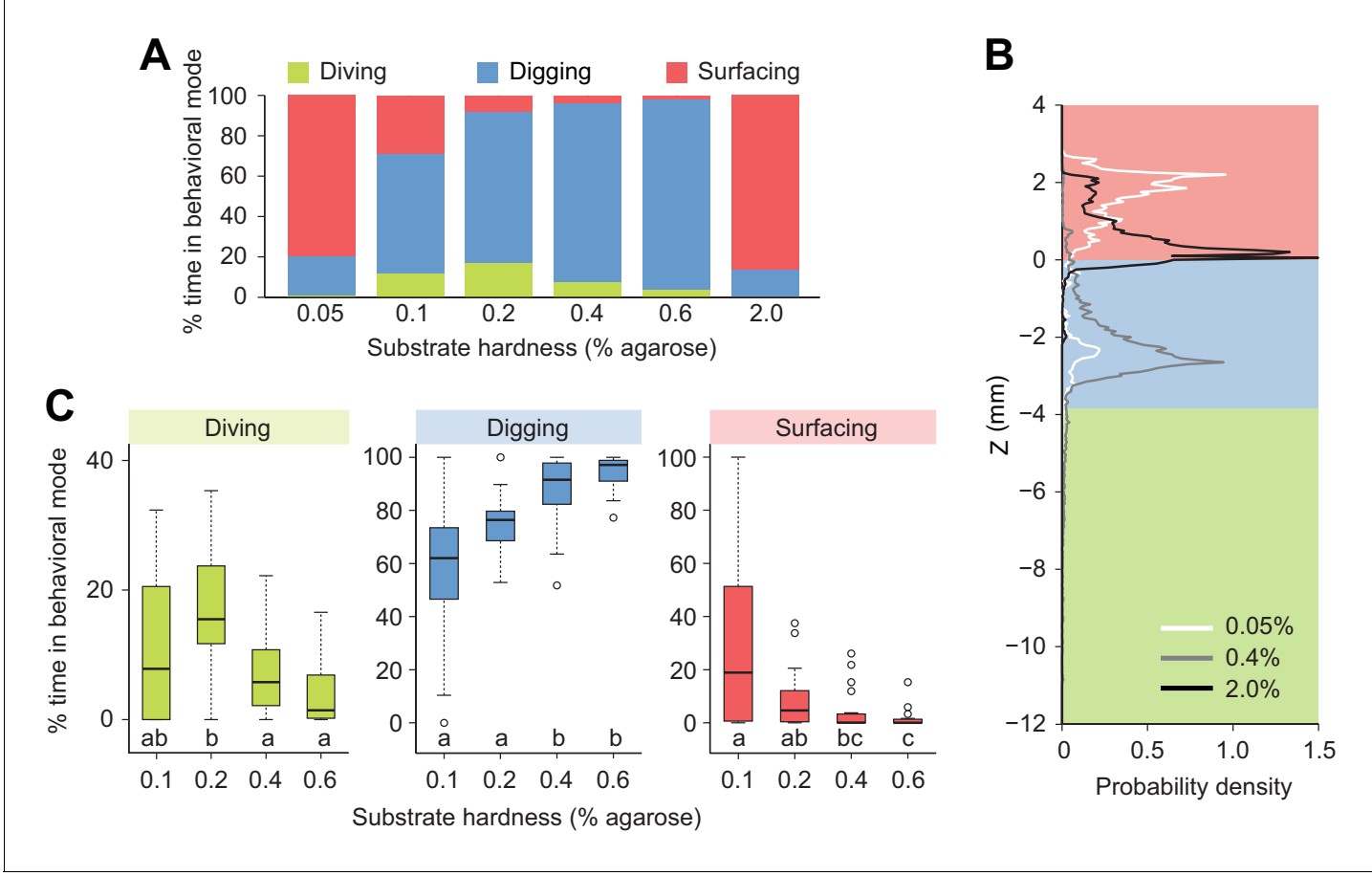

**Figure 3.** Modulation of exploratory behavior by the hardness of the substrate. (**A**) Percentage of time spent in each of the three elementary behavioral modes: diving (green), digging (blue) and surfacing (red) of larvae foraging in agarose gels of different hardness. (**B**) Probability density functions of larval centroid depth (Z) in very soft (0.05%), moderately stiff (0.4%) and solid-like (2.0%) agarose gels. (**C**) Boxplots of percentage of time in behavioral modes in intermediate range of the substrate hardness, 0.1–0.6% agarose. Samples with different letters indicate significantly different medians (Kruskal-Wallis test followed by pairwise Wilcoxon rank-sum test with Bonferroni correction, p<0.05, n = 24 trials). More information about the statistics is given in *Supplementary file 1*.

DOI: https://doi.org/10.7554/eLife.27057.006

The following figure supplements are available for figure 3:

**Figure supplement 1.** Diving in soft substrates involves the risk of drowning.

DOI: https://doi.org/10.7554/eLife.27057.007

**Figure supplement 2.** Quantification of single dives in substrates of different hardness.

DOI: https://doi.org/10.7554/eLife.27057.008

(*Grant et al., 2012*). In low-percentage agarose gels (0.05% w/v), surfacing was the dominant behavioral mode in *D. melanogaster* (*Figure 3A*). Nearly no diving was observed in 0.05% agarose gels (*Figure 3A*), suggesting that motion through peristalsis is inefficient in soft hydrogels (*Video 1*). Compared to substrate of intermediate hardness (0.4%), the probability distribution of the larval centroid was displaced toward the air chamber in 0.05% agarose gels (*Figure 3B*). The peak of the PDF corresponding to 0.05% agarose gels coincided with the top of the air chamber, indicating that *D. melanogaster* larvae actively avoided soft substrates after having experienced their textures (*Video 1*). Individuals that failed to return to the surface stopped moving after a few minutes. Given that inactive larvae were found to have drowned (*Figure 3—figure supplement 1*), immersed animals that stopped being active were not included in the analysis. In the other extreme condition of hardness, larvae were tested in hard media of 2% agarose (*Video 2*). In this condition, larvae rested at the surface of the gel during ~80% of the trial with short periods of digging (*Figure 3A*). This trend is reflected in the shift of the PDF of larval positions toward the

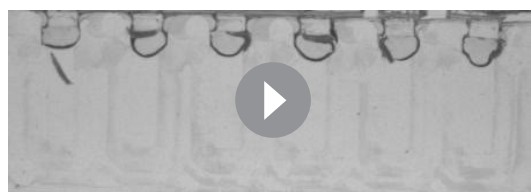

**Video 1.** Exploratory behavior of *D. melanogaster* larvae in dig-and-dive chambers containing 0.05% plain agarose gel. Replay speed is 10 times faster than the original behavior.
DOI: https://doi.org/10.7554/eLife.27057.009

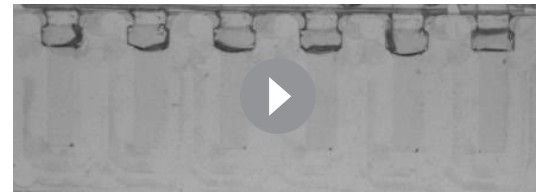

**Video 2.** Exploratory behavior of *D. melanogaster* larvae in dig-and-dive chambers containing 2.0% plain agarose gel. Replay speed is 10 times faster than the original behavior.
DOI: https://doi.org/10.7554/eLife.27057.010

air chamber (*Figure 3B*). For densities equal to or higher than 3% agarose, *D. melanogaster* larvae could neither dig nor dive into the substrate (*Louis et al., 2008*; *Apostolopoulou et al., 2014*). For intermediate gel hardness, larvae spent variable fractions of time in the three main behavioral modes: surfacing, digging and diving (*Figure 3A*). The largest percentage of time spent in dive mode was observed in gels of intermediate hardness (0.2% agarose). For media with an agarose percentage higher than 0.2%, the median duration of individual dives decreased with the hardness of the gel, and so did the maximum dive depth (*Figure 3—figure supplement 2*). In summary, we found that *D. melanogaster* larvae spent most of their time at the surface when the environment was either too soft or too hard. For media with intermediate textures, the balance between digging and diving was influenced by the stiffness of the gel: diving tended to be suppressed as the hardness of the substrate increased (*Figure 3*). The difficulty of *D. melanogaster* larvae to dig into hard substrates probably explains their preference for the internal soft layers of tomato slices (*Figure 1—figure supplement 1*).

## Influence of respiration on diving behavior

The respiration system of the *Drosophila* larva is composed of two parallel tracheae that terminate in anterior and posterior spiracles enabling gas exchange with the external environment (*Figure 4A*) (*Manning and Krasnow, 1993*). In the tracheae of the larva, longitudinal transport of oxygen takes place by means of passive diffusion (*Krogh, 1920*). Upon submersion into liquid, the extremities of the anterior and the posterior spiracles are closed by hydrophobic hairs (*Manning and Krasnow, 1993*). As larvae feed head-down with their posterior end in contact with the surface (*Figure 2B*, right panel), the two posterior spiracles can be viewed as 'snorkeling' devices that enable larvae to stay oxygenated while digging into the medium. During digging and diving, the anterior spiracles are immersed and retracted inside the head (*Manning and Krasnow, 1993*) to prevent the flow of external liquid into the tracheae. During digging, gas exchange is still achieved through the posterior spiracles, which allows larvae to stay oxygenated through their posterior spiracles without having to fully surface during the entire duration of an experiment. By contrast, diving is limited by the lack of oxygenation following the loss of contact of the spiracles with the surface. When larvae are unable to return to the surface to re-oxygenate, movements stop. In *Figure 3—figure supplement 1*, we established that the lack of movement coincides with drowning.

To assess the influence of oxygenation on diving behavior, we artificially obstructed the posterior spiracles with thermoplastic glue (*Figure 4A*, bottom panel). *D. melanogaster* larvae with blocked posterior spiracles were unharmed and capable of breathing through their anterior spiracles. We controlled that their locomotor activity (average speed) was not strongly impaired during motion on two-dimensional (2D) agarose surfaces (*Figure 4—figure supplement 1*). In addition, we verified the absence of differences in locomotor patterns (body postures) between larvae with intact and blocked posterior spiracles (*Videos 3* and *4*). In spite of their normal locomotor behavior, larvae with blocked posterior spiracles displayed dramatic differences in the organization of their exploratory behavior in the dig-and-dive assay. Deficient respiration produced a strong suppression of digging and diving (*Figure 4B and C*). Upon block of the posterior spiracles, the percentage of time spent digging and diving was significantly decreased whereas the time spent surfacing increased (*Figure 4D*). Based on these results, we speculate that exploratory dives induce hypoxic stress that

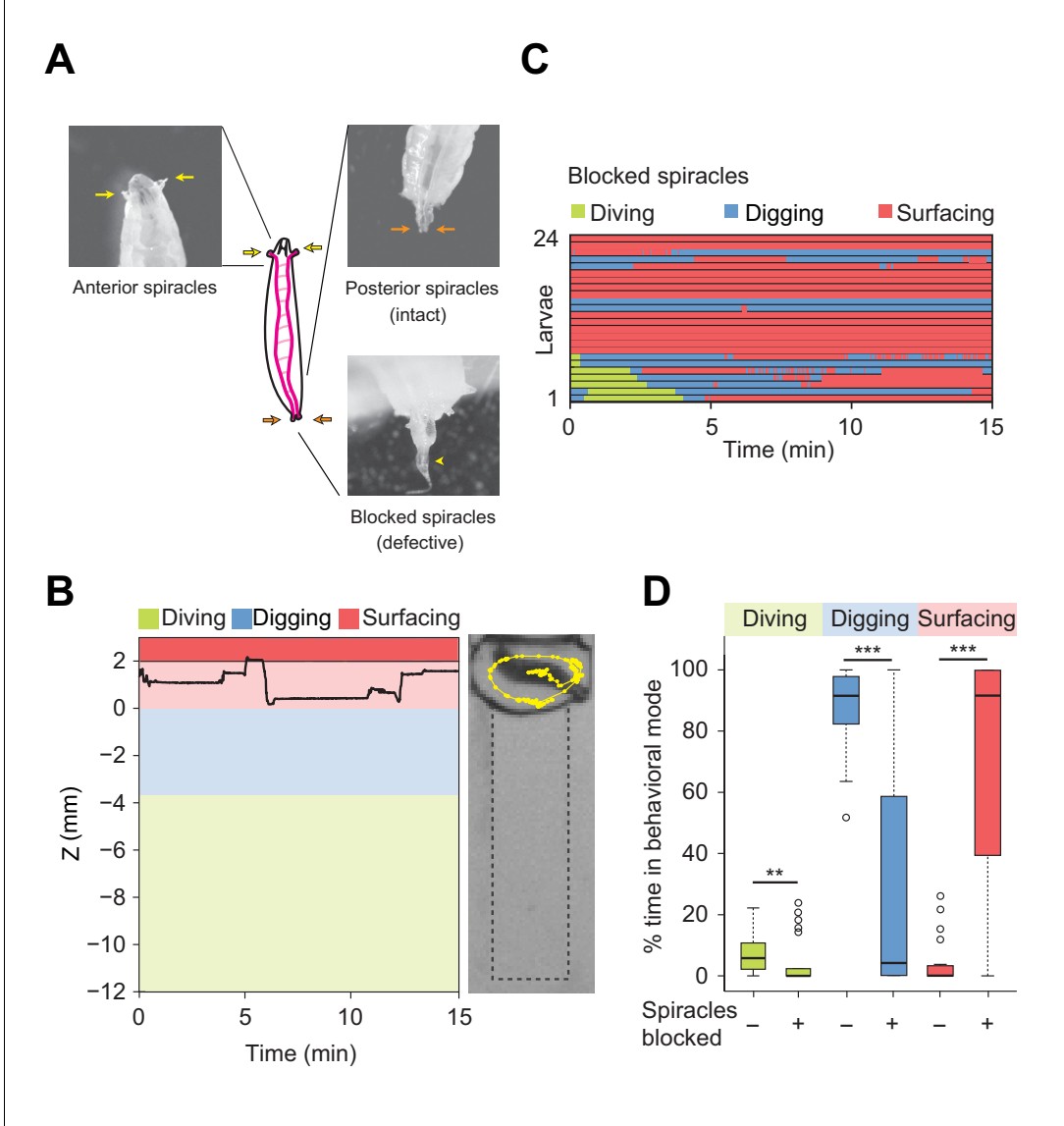

**Figure 4.** Modulation of exploratory behavior by respiration. (**A**) Schematic of the larval tracheal system and close-up view of spiracles. The magenta lines in the diagram illustrate the two dorsal main trunks of the trachea (parallel to the longitudinal body axis) and their lateral branches (perpendicular to the longitudinal body axis). Top panel: the arrows indicate two anterior (yellow) and two posterior (orange) spiracles. Bottom panel: the arrowhead indicates blocked posterior spiracles with thermoplastic adhesives. (**B**) Left panel: time course of the centroid depth (Z) of a representative of spiracle-blocked larva during a 15 min trial. Right panel: picture of the assay overlaid with a complete 15 min trajectory (dots highlight positions every 1 s). (**C**) Ethograms over time of larvae with blocked spiracles (n = 24 trials). Each row of the ethogram corresponds to a different animal. The time course of the behavioral state of a larva is represented according to the color code at the top of the panel. Trials were sorted by increasing total dive times. Average numbers of dives per trial are 4.1 ± 3.2 and 0.3 ± 0.6 (mean ± s.d.), respectively. (**D**) Percentage of time in behavioral modes for larvae with intact and blocked spiracles (Wilcoxon rank-sum test, **p<0.01 and ***p<0.001). More information about the statistics is given in *Supplementary file 1*.
DOI: https://doi.org/10.7554/eLife.27057.011

The following figure supplement is available for figure 4:

**Figure supplement 1.** Locomotor activity of larvae with intact and blocked spiracles.
DOI: https://doi.org/10.7554/eLife.27057.012

forces larvae to stay at the surface. In line with the active avoidance of very soft substrates that pose a risk of drowning during dives (*Figure 3* and *Figure 3—figure supplement 1*), diving can be viewed as a form of risk-taking behavior, which is part of the larval foraging strategy.

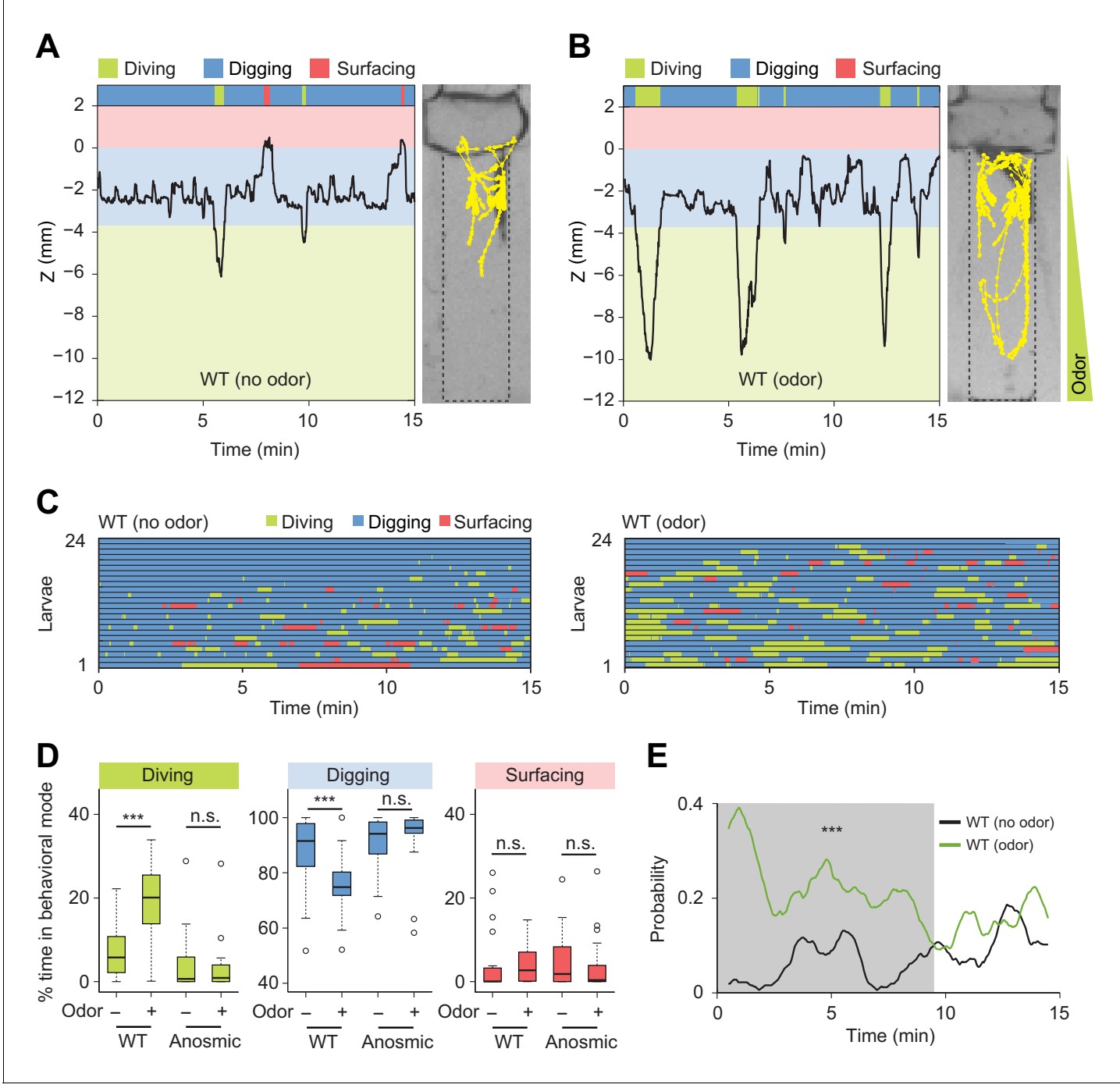

**Figure 5.** Induction of exploratory behavior by an attractive odor (ethyl acetate). (A) Representative time course of centroid depth (Z) of a wild-type (WT) larva and corresponding trajectory in 0.4% agarose gel in absence of odor (dots highlight positions every 1 s). (B) Same as in presence of the odor (ethyl acetate). The larva reached the bottom of the agarose chamber when its centroid was at the depth close to −10 mm. (C) Ethograms for wild-type larvae tested in absence and in presence of the odor (n = 24 trials). The ethogram shown in panel C for the absence of odor is the same as that presented in *Figure 2D*. Average numbers of dives per trial are 4.1 ± 3.2 and 5.3 ± 2.5 (mean ± s.d.), respectively. Trials were sorted by increasing total dive times. (D) Percentages of total time spent in behavioral modes for wild-type and anosmic ($Orco^{-/-}$) larvae with olfactory impairment tested in an agarose gel without/with the odor (Wilcoxon rank-sum test, n = 22–24 trials). (E) Time course of probability of diving for wild-type larvae without (black) and with (green) the odor. A sliding window of 1 min was applied. The gray background represents the period during which the likelihood of diving in the agarose with the odor is higher than that without the odor (Kolmogorov-Smirnov test). For (D) and (E), significance levels of the statistics are indicated as ***p<0.001 and n.s. stands for not significant. More information about the statistics is given in *Supplementary file 1*.
DOI: https://doi.org/10.7554/eLife.27057.015

*Figure 5 continued on next page*

*Figure 5 continued*

The following figure supplements are available for figure 5:

**Figure supplement 1.** Vertical gradient of chemical concentration.
DOI: https://doi.org/10.7554/eLife.27057.016
**Figure supplement 2.** Detailed analysis of diving behavior of wild-type larva in absence and in presence of an attractive odor (ethyl acetate).
DOI: https://doi.org/10.7554/eLife.27057.017
**Figure supplement 3.** Diving is impaired by the detection of yeast mixed in the agarose substrate.
DOI: https://doi.org/10.7554/eLife.27057.018
**Figure supplement 4.** Diving is significantly reduced by the perception of sugar mixed in the agarose substrate.
DOI: https://doi.org/10.7554/eLife.27057.019
**Figure supplement 5.** Feeding behavior is mostly associated with digging.
DOI: https://doi.org/10.7554/eLife.27057.020

## Olfactory stimulation promotes exploratory behavior

Odorant molecules detected in gaseous phase direct orientation behavior in larvae crawling on 2D agarose surfaces (*Gomez-Marin and Louis, 2012*). To the best of our knowledge, the role of olfaction has never been tested in conditions relevant to the natural habitat of larvae, e.g. the pulp of fermenting fruits. Using the dig-and-dive assay, we asked whether larvae engage in exploratory dives when appetitive odors are embedded in the medium. We injected a small quantity of odor at the bottom of the diving chamber. Ethyl acetate (EtA, 40 mM of aqueous solution) was used due to its high solubility in water (82.2 g/L at 22°C) and the strength of the chemotactic behavior it elicits in gaseous phase (*Kreher et al., 2008*). *D. melanogaster* larvae were tested in 0.4% agarose gels that included a stable odor gradient pointing from the surface down to the bottom of the assay (*Figure 5* and *Figure 5—figure supplement 1*).

In the absence of odor, *D. melanogaster* larvae displayed infrequent dives that were short-lived and shallow (*Figure 5A and C*, and *Video 5*). By contrast, the presence of the odor induced a significant prolongation of individual dives together with an increase in dive frequency and dive depth (*Figure 5B and C*, *Figure 5—figure supplement 2*, and *Video 6*). As a result, the percentage of time spent in diving mode increased in presence of the odor (*Figure 5D*). The increase in dive time was coupled to a marginal increase in the number of dives (average number of dives: 4.1 ± 3.2 in absence of odor and 5.3 ± 2.5 in presence of odor). While larvae did not dive into the medium directly upon their introduction in a chamber devoid of odor, the presence of the odor promoted diving shortly after the beginning of the trial (*Figure 5C*, right panel). Therefore, olfactory

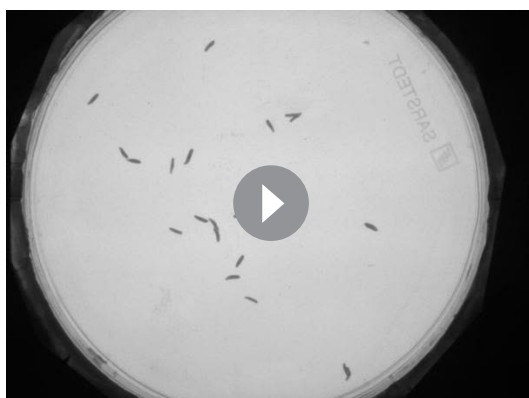

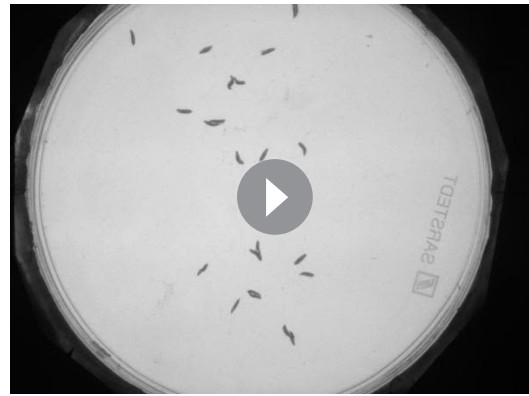

**Video 3.** Crawling behavior of intact *D. melanogaster* larvae on a 2% plain agarose gel. Replay speed is real time.
DOI: https://doi.org/10.7554/eLife.27057.013

**Video 4.** Crawling behavior of *D. melanogaster* larvae with blocked posterior spiracles. Larvae were tested on a 2% plain agarose gel similar to *Video 3*. Replay speed is real time.
DOI: https://doi.org/10.7554/eLife.27057.014

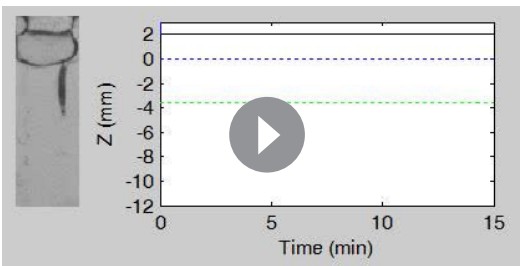

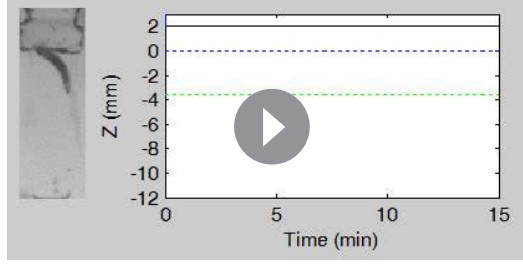

**Video 5.** Exploratory behavior of *D. melanogaster* larva in a dig-and-dive chamber containing 0.4% plain agarose gel. Replay speed is 10 times faster than the original behavior.
DOI: https://doi.org/10.7554/eLife.27057.021

**Video 6.** *D. melanogaster* larva in a dig-and-dive chamber containing 0.4% plain agarose gel with a stable gradient of ethyl acetate (EtA). The presence of the odor induces more frequent, longer, and deeper dives. Replay speed is 10 times faster than the original behavior.
DOI: https://doi.org/10.7554/eLife.27057.022

stimulation had a synchronizing effect on the behavior of larvae across trials. This increase in the probability of diving was sustained for several minutes compared to the condition without odor (*Figure 5E*).

To further characterize the effects of the attractive odor on the control of the dive dynamics, we inspected the cumulative distribution functions (CDF) of the maximum dive depths and the single dive times (*Figure 5—figure supplement 2A and B*). These two functions showed that larvae dived deeper and longer in the presence of the odor. We found a positive correlation between diving time and the maximum depth of a dive. The durations of individual dives increased with the maximum dive depth. Moreover, the depths of individual dives increased in presence of the odor (*Figure 5—figure supplement 2C*). This result suggests that *D. melanogaster* larvae spent more time searching for the odor source at the bottom of the chamber. On the other hand, the presence of the odor led to a non-significant increase in dive frequency while the dive speed remained unchanged (*Figure 5—figure supplement 2D and E*). We observed no correlation between the duration of a dive and the duration of surfacing that followed diving (*Figure 5—figure supplement 2F*). This result argues that most dives did not reach their aerobic limit (*Kooyman et al., 1980*; *Boyd, 1997*).

In short, *D. melanogaster* showed an attractive response to the food odor, ethyl acetate, by increasing the time larvae spent in diving mode with a synchronization of their first dive at the beginning of the trial. To examine whether these behavioral changes were mediated by the olfactory system, we tested the dig-and-dive behavior of larvae with genetically silenced olfactory sensory neurons (*Larsson et al., 2004*). Larvae with impaired olfactory function did not respond to the presence of the liquid-borne odor (*Figure 5D*). This established that the behavioral response elicited by liquid-borne odors is not predominantly mediated by the gustatory system. Taken together, the previous results indicate that larvae can make use of the detection of odors in liquid phase to direct explorative dives toward a potential food source. In agreement with this idea, diving is nearly fully impaired when larvae detect the presence of natural food (yeast) mixed with the substrate (*Figure 5—figure supplement 3*). The addition of sugar (fructose) alone to the substrate was found to have the same blocking effect on diving (*Figure 5—figure supplement 4*). Moreover, sugar promoted feeding behavior through repeated mouth hook contractions during digging but not during diving (*Figure 5—figure supplement 5*). The basal rate of mouth hook contraction observed during diving is part of the locomotor program that permits larvae to move in the agarose substrate. In conclusion, diving can be viewed as an exploratory food-search behavior that can be driven by olfactory cues but that is largely uncoupled from active feeding.

## Ecological differences in dig-and-dive behavior between *Drosophila melanogaster* and *Drosophila suzukii*

In nature, *Drosophila* species experience a wide variety of food resources whose properties differ in stiffness and nutrient composition (*Hansson and Stensmyr, 2011*). Digging and diving behaviors between different species are expected to partially reflect an adaption to differences in food resources and ecological niches. To investigate the relationship between the propensity of larvae to

**Figure 6.** Effects of substrate hardness and respiration on the exploratory behavior of *Drosophila suzukii* larvae. (A) Percentages of time in the three behavioral modes observed in gels with different percentages of agarose. (B) Ethogram over time of intact larvae of the *Drosophila suzukii* group (*n* = 24 trials). Average number of dives per trial is 1.9 ± 1.5 (mean ± s.d.). (C) Duration of single dives of *D. melanogaster* and *D. suzukii* larvae with intact spiracles (Wilcoxon rank-sum test, *n* = 98 and 45 dives, respectively). (D) Percentages of total time spent in diving mode of *D. melanogaster* with intact spiracles compared to *D. suzukii* larvae with intact and blocked posterior spiracles (Wilcoxon rank-sum test). (E) Ethogram of the *D. suzukii* larvae with blocked posterior spiracles (*n* = 23 trials). Trials were sorted by increasing total dive times. Average number of dives per trial is 1.0 ± 0.9 (mean ± s. d.). For (C) and (D), significance levels of the statistics are indicated as ***p<0.001. Results shown in panels (C) and (D) for *D. melanogaster* arise from the same dataset as *Figure 4*. For more information about the statistics, see *Supplementary file 1*.

DOI: https://doi.org/10.7554/eLife.27057.023

The following figure supplement is available for figure 6:

**Figure supplement 1.** *D. suzukii* larvae display a preference for soft substrates.
DOI: https://doi.org/10.7554/eLife.27057.024

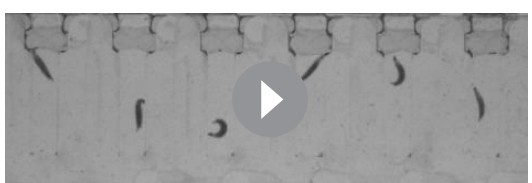

**Video 7.** Exploratory behavior of *D. suzukii* larvae in dig-and-dive chambers filled with a 0.05% plain agarose gel. Replay speed is 10 times faster than the original behavior.
DOI: https://doi.org/10.7554/eLife.27057.025

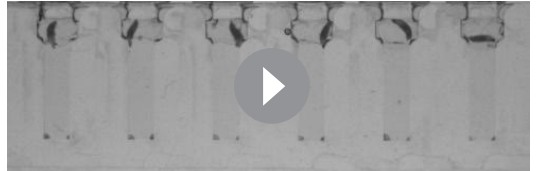

**Video 8.** Exploratory behavior of *D. suzukii* larvae in dig-and-dive chambers filled with a 2% plain agarose gel. Replay speed is 10 times faster than the original behavior.
DOI: https://doi.org/10.7554/eLife.27057.026

engage in exploratory dives and the hardness of food resources, we chose *Drosophila suzukii* as a second species. Unlike *D. melanogaster*, *D. suzukii* preferentially targets fresh (hard) fruits over (soft) rotten fruits for egg laying (*Karageorgi et al., 2017*). We used the dig-and-dive assay to characterize the organization of digging and diving behaviors in *D. suzukii* larvae (*Figure 6*). In soft media (0.05%) in which peristaltic locomotion was mostly impaired in *D. melanogaster* (*Figure 3A* and *Video 1*), *D. suzukii* larvae did not refrain from diving (*Figure 6A* and *Video 7*). In media with intermediate hardness (0.4%), the fraction of time spent diving was significantly larger for *D. suzukii* than for *D. melanogaster* (*Figure 6D*). As observed in the ethogram of *Figure 6B*, the average number of dives per trial was lower for *D. suzukii* than for *D. melanogaster* (1.9 ± 1.5 and 4.1 ± 3.2, respectively). The increase of the fraction of time spent in dive mode resulted from a tenfold increase in the duration of individual dives (*Figure 6C*), reflecting the higher tolerance of *D. suzukii* larvae to hypoxia. This hypothesis was corroborated by the enhanced robustness of *D. suzukii* to hypoxic stress induced by a block of its posterior spiracles (*Figure 6D*).

Overall, *D. suzukii* dived deeper than *D. melanogaster* (black traces, *Figure 8—figure supplement 1*) and *D. suzukii* initiated diving earlier during the trial (*Figure 6B* to be compared with *Figure 5C*, left panel). Like in *D. melanogaster*, the increase in surfacing time that followed spiracle block in *D. suzukii* highlights the reliance of digging and diving on respiration through the posterior spiracles (*Figure 6E*). Although blocking the spiracles of *D. suzukii* reduced the average number of dives per trial (1.9 ± 1.5 versus 1.0 ± 0.9, respectively), it did not severely decrease the percentage of time spent in dive mode (*Figure 6D*). In hard media (2%), *D. suzukii* larvae showed a fourfold increase in cumulated digging time compared to *D. melanogaster*, strengthening the idea that the ability of *D. suzukii* to grow on fresh fruits is coupled with an adaptation to dig and dive into hard substrates (*Video 8* and *Figure 6A* for *D. suzukii* to be compared to *Figure 3A* for *D. melanogaster*). When given a choice, *D. suzukii* larvae nonetheless displayed a preference for soft substrates (*Figure 6—figure supplement 1*) — a phenomenon also observed for feeding behavior at the adult stage (*Karageorgi et al., 2017*).

Next, we characterized the modulatory effect of an attractive odor on the dig-and-dive behavior of *D. suzukii*. Since the olfactory response profile of this species has not been studied as in *D. melanogaster* (*Fishilevich et al., 2005*; *Mathew et al., 2013*), we selected the same attractive food odor used to test the behavior of *D. melanogaster*: ethyl acetate (*Figure 5*). First, we established the attractiveness of ethyl acetate to *D. suzukii* in a standard chemotaxis assay in which the odor was delivered in gaseous phase (*Figure 7—figure supplement 1*). When presented in the dig-and-dive assay, ethyl acetate led to a significant reduction in total dive time in *D. suzukii* (*Figure 7B*) with no dramatic decrease in the duration of individual dives (median times of 1.1 and 0.8 min for the no-odor and the odor conditions, respectively; Wilcoxon rank-sum test p=0.24). The decrease in total dive time could result from the transient nature of the attractive response and/or a gradual aversion to the odor. We examined both possibilities by characterizing the change in diving behavior induced by the odor. While a decrease in the number of dives is expected for aversive responses, we found no obvious reduction in the average number of dives with and without odor (2.0 ± 1.2 and 1.9 ± 1.5, respectively). This latter result is consistent with the observation that ethyl acetate is an odor attractive to *D. suzukii* in gaseous phase (*Figure 7—figure supplement 1*).

We then considered the depth to which each species dived. Even in the absence of odor, *D. suzukii* frequently reached the bottom of the chamber. By contrast, *D. melanogaster* performed shallower dives (the maximum dive depth for *D. suzukii* and *D. melanogaster* is 7.1 ± 3.4 mm and 5.2 ± 1.6 mm, respectively). In presence of ethyl acetate, *D. melanogaster* dived as deep as *D. suzukii* (6.6 ± 2.7 mm and 6.6 ± 3.0 mm, respectively). Since the odor affected neither the time spent at the surface (*Figure 7B*) nor the average dive depth, we concluded that ethyl acetate is unlikely to act as a repellent in *D. suzukii*. Although the odor synchronized diving at the beginning of the trial (*Figure 7C*), the increase in dive probability was only transient compared to the sustained stimulation of diving in *D. melanogaster* (*Figure 5E*). Taken together, our results indicate that *D. suzukii* undergo a transient attractive response to the food-odor ethyl acetate in the dig-and-dive assay. The transient nature of the response in *D. suzukii* contrasts with the persistent attraction the odor elicited in *D. melanogaster*.

Finally, we examined the modulatory effect of a repulsive odor, geosmin — an earthy volatile chemical produced by fungi and bacteria. Geosmin has been shown to elicit avoidance responses in adult flies of *D. melanogaster* and *D. suzukii* (*Stensmyr et al., 2012*). Using the dig-and-dive assay,

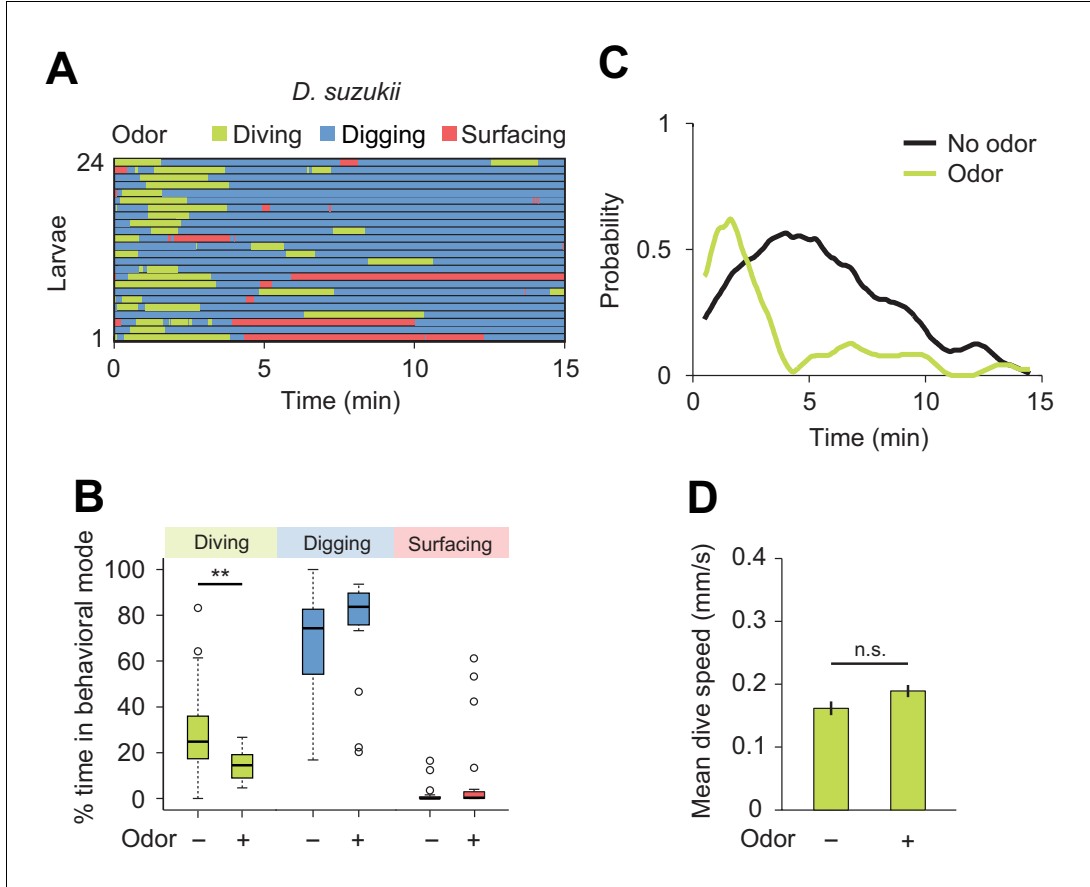

**Figure 7.** Behavioral responses of *Drosophila suzukii* larvae to an attractive odor (ethyl acetate). (A) Ethograms in assay filled with agarose where 1 μL of odor (40 mM ethyl acetate, EtA) injected at 11 mm depth below the surface of the gel (n = 24 trials). Average number of dives per trial is 2.0 ± 1.1 (mean ± s.d.). (B) Percentages of total time spent in behavioral modes (Wilcoxon rank-sum test, n = 24 trials). (C) Time course of probability of diving with total number of larvae without (black) and with (green) the odor. A sliding window of 1 min was applied. (D) Mean dive speed of larvae (means ± s.e.m., n = 45 and n = 48 dives, Student's *t*-test). For (B) and (D), significance levels of the statistics are indicated as **p<0.01 and n.s. for not significant. More information about the statistics is given in *Supplementary file 1*.

DOI: https://doi.org/10.7554/eLife.27057.027

The following figure supplement is available for figure 7:

**Figure supplement 1.** *Drosophila suzukii* larvae are attracted by the airborne food-odor ethyl acetate.

DOI: https://doi.org/10.7554/eLife.27057.028

we tested whether geosmin (0.5 mM) induced aversive behavior in larvae. Like for the attractive odor, a small quantity of geosmin was injected near the bottom of the agarose gel. For both *D. melanogaster* and *D. suzukii*, detection of geosmin affected the dynamics of the dig-and-dive behavior. However, several aspects of the behavioral modulation were species-specific. For *D. melanogaster*, geosmin did not induce a significant change in the fraction of time spent diving, but it significantly increased surfacing time — a trend characteristic of aversion (*Figure 8B and C*). While surfacing larvae avoided contact with the gel, corroborating the idea that geosmin acted as a repellent in *D. melanogaster* (*Figure 8—figure supplement 1A*). It was nonetheless observed that geosmin had a synchronizing effect on the first dive at the beginning of the trial (*Figure 8C*). Thus, *D. melanogaster* showed a transient attractive response at the beginning of the trial that turned into prolonged aversion. For *D. suzukii*, geosmin produced a significant decrease in the fraction of time spent diving (*Figure 8E*) coupled with a significant increase in surfacing time. Geosmin suppressed diving shortly after the beginning of the trial (left panel of *Figure 8F*). It also decreased the probability of diving deeper and increased the probability of staying above the surface of the gel (*Figure 8—figure*

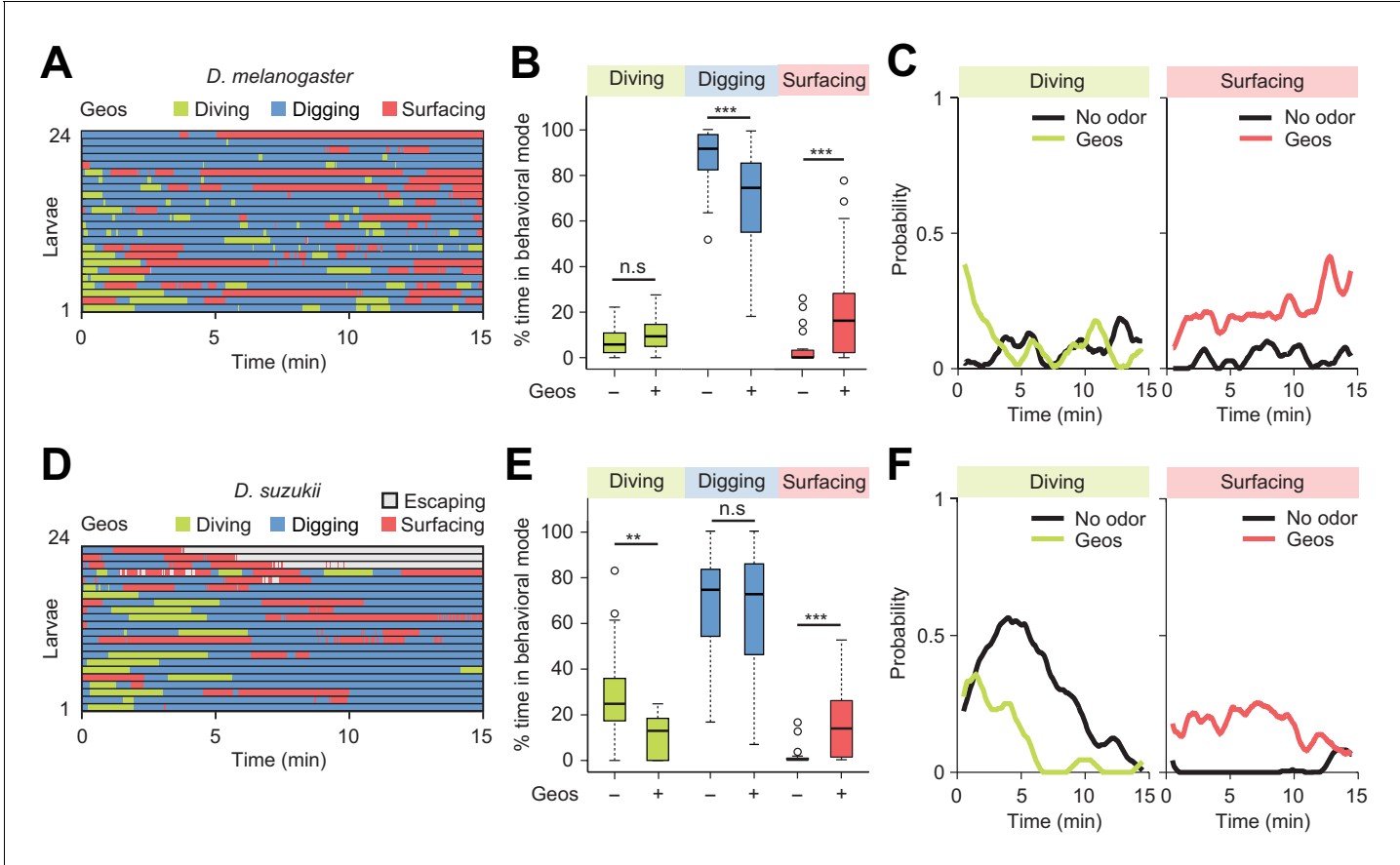

**Figure 8.** Behavioral responses to an aversive odor (geosmin) in two different Drosophila species. (A, D) Ethograms of *D. melanogaster* and *D. suzukii* larvae in the presence of 1 μL of 0.5 mM geosmin (Geos) injected at a depth of 11 mm below the surface of the gel (*n* = 24 and *n* = 22 trials). Average numbers of dives per trial are 3.7 ± 3.1 and 1.2 ± 1.1 (mean ± s.d.). (B, E) Percentages of total time spent in behavioral modes for different species. *D. melanogaster* in (B) and *D. suzukii* in (E) (Wilcoxon rank-sum test). (C, F) Time courses of probability of diving and surfacing with total number of larvae without and with the odor. *D. melanogaster* in (C) and *D. suzukii* in (F). A sliding window of 1 min was applied. Significance levels of the statistics are indicated as \*\*$p<0.01$, \*\*\*$p<0.001$, and n.s. for not significant. For more information about the statistics, see *Supplementary file 1*.

DOI: https://doi.org/10.7554/eLife.27057.029

The following figure supplements are available for figure 8:

**Figure supplement 1.** Repulsive responses induced by stimulations with geosmin in the dig-and-dive assay.

DOI: https://doi.org/10.7554/eLife.27057.030

**Figure supplement 2.** Response of *Drosophila suzukii* larvae to DEET in the dig-and-dive assay.

DOI: https://doi.org/10.7554/eLife.27057.031

*supplement 1B*). The ethogram of *D. suzukii* illustrates the emergence of a fourth behavioral mode: escaping out of the arena through the air channels (light gray traces in *Figure 8D*). Considering the difference between the diameter of the air channels (~0.2 mm) through which larvae attempted to escape and the diameter of the larval body (~1 mm), escaping must be a behavior that requires substantial energy. Upon stimulation with geosmin, a quarter of the larvae escaped the assay while none did so in absence of the chemical. As shown in *Figure 8—figure supplement 2*, similar escape responses were observed for *D. suzukii* exposed to the common insect repellent DEET (N,N-Diethyl-meta-toluamide, *Brown and Hebert, 1997*; *Ditzen et al., 2008*; *Katz et al., 2008*). Based on these observations, we conclude that geosmin acts as a repellent in the *D. melanogaster* and *D. suzukii* species: it promotes exit and escape from the substrate.

## Discussion

How animals regulate food-search behavior and exposure to uncertainty related to feeding is an intriguing, yet poorly understood problem. Here we present a new assay to study how the integration of sensory cues and internal states modulate the dynamics of exploratory behaviors in the *Drosophila* larva. While larval orientation behavior has been extensively examined on planar surfaces in response to stable airborne odor gradients (*Louis et al., 2008*; *Gershow et al., 2012*), larvae live in complex tridimensional hydrogel structures formed by fruits and decaying organic materials. We developed a novel assay called *dig-and-dive* to monitor the exploratory behavior of larvae in diving chambers filled with agarose (*Figure 2*). Like in previous analysis of crawling behavior on planar surfaces (*Green et al., 1983*), we defined elementary modes of behavior. Those include: (1) surfacing when most of the body laid above the surface of the hydrogel, (2) digging when the body was immersed into the agarose but the posterior spiracles remained in contact with the surface, (3) diving when the larva broke contact with the surface. We showed that active feeding is associated with digging but not diving (*Figure 5—figure supplement 5*). In addition, we defined (4) escaping as the aversive response displayed by larvae that attempted to leave the assay upon detection of repulsive chemicals.

*Drosophila* larvae move through peristaltic crawling (*Berrigan and Pepin, 1995*). Although the biomechanics of peristalsis is not fully understood, locomotion results from the interactions of the body with the environment (*Omori et al., 2009*). The mechanical properties of the substrate surrounding the larva are therefore expected to affect the efficiency of its motion. Accordingly, *D. melanogaster* larvae were highly sensitive to the hardness of the hydrogel used in the dig-and-dive assay. Diving was impaired in hard media with an agarose percentage superior to 2% (*Figure 3A* and *Video 2*). By contrast, diving was observed in softer gels with an agarose percentage that ranged between 0.1% and 0.6%. This result is consistent with the behavior of *D. melanogaster* in tomato slices where larvae demonstrated a marked preference to dig into the soft part of the fruit (hardness inferior to 1% agarose), while avoiding the harder part of the fruit (hardness superior to 2% agarose, *Grant et al., 2012*; *Li et al., 2012*) (*Figure 1* and *Figure 1—figure supplement 1*). In very soft gels with an agarose percentage of 0.05%, the insufficient friction precluded efficient motion and posed the risk of drowning (*Figure 3A* and *Figure 3—figure supplement 1*). *D. melanogaster* larvae that attempted to dive in soft gels refrained from initiating subsequent dives (*Video 1*), which suggests that individual larvae were capable of assessing the risk of drowning in very soft gels.

Compared to most other *Drosophila* species that feed preferentially on ripe or damaged fruits having fallen from the tree, *Drosophila suzukii* targets fresh fruits to lay their eggs (*Keesey et al., 2015*; *Karageorgi et al., 2017*). Females of *D. suzukii* oviposit below the skin of berries and cherries. When *D. suzukii* larvae hatch, they must feed and grow in conditions that are stiffer than *D. melanogaster*. We therefore hypothesized about the existence of species-specific differences in the dig-and-dive behavior of *D. suzukii* and *D. melanogaster*. We found that *D. suzukii* moved more efficiently than *D. melanogaster* in soft substrates. In addition, *D. suzukii* were capable of diving into hard media impenetrable to *D. melanogaster* (*Video 8* and compare the 2% agarose condition in *Figure 3A* and *Figure 6A*). This hints at a possible specialization of the locomotor program of *D. suzukii* to permit digging in stiffer conditions. While the organization of peristalsis does not appear to be grossly different between both species, *D. suzukii* larvae are slightly larger than *D. melanogaster* (*Walsh et al., 2011*). Moreover, *D. suzukii* larvae implement longer dives than *D. melanogaster* (*Figure 6C*). This behavioral difference might reflect a higher tolerance of *D. suzukii* to oxygen deficits (hypoxia). Consistent with this idea, obstructing the posterior spiracles of *D. suzukii* larvae did not fully impair diving in *D. suzukii* (*Figure 6D and E*). In future work, it will be interesting to define whether the enhanced ability of *D. suzukii* to dig and dive into hard media results from a specialization of specific body parts such as the mouth hooks or an adaptation of the locomotor program producing thrust during dives. As was observed in adult flies (*Karageorgi et al., 2017*), the ability of *D. suzukii* larvae to cope with stiffer substrates does not translate into a preference to feed on hard versus soft media (*Figure 6—figure supplement 1*).

Peristaltic locomotion consumes large amounts of energy produced through respiration (*Berrigan and Lighton, 1993*). As oxygen reserves are gradually consumed during apneic dives, larvae turn hypoxic and eventually return to the surface of the gel to re-oxygenate. Failure to re-

oxygenate leads to complete immobility, which was shown to be associated with death through drowning (*Figure 3—figure supplement 1*). Diving can be viewed as a primitive form of risk-taking behavior directed by a balance between exploitation (surfacing and feeding through digging) and exploration (apneic dives). The importance of respiration in the control of diving was demonstrated here by subjecting larvae to non-lethal hypoxic stress (mechanical block of posterior spiracles, *Figures 4C* and *6E*). Impairment of respiration reduced the probability of engaging in digging and diving in both species. *D. suzukii* nonetheless displayed a higher resistance to hypoxia than *D. melanogaster* (*Figure 6D*). The alternation between sequences of apneic dives and surfacing is reminiscent of food search behaviors in sea mammals. Whales, seals and turtles seek food during dives interspaced with resting phases at the surface (*Boyd, 1997*; *Tyack et al., 2006*; *Houghton et al., 2008*). The lack of correlation between the duration of single dives and the subsequent resting phase suggests that most sea mammals perform aerobic dives (*Boyd, 1997*; *Hooker and Baird, 1999*). The same conclusion can be drawn for the *D. melanogaster* larva (*Figure 5—figure supplement 2*). How the duration of a dive is physiologically controlled by hypoxia remains to be elucidated.

The importance of olfactory cues has been established in the food-search behavior of terrestrial insects, reptiles and mammals (*Fraenkel and Gunn, 1961*) as well as birds, crustaceans and fishes (*Guilford et al., 1998*; *DeBose and Nevitt, 2008*; *Nevitt et al., 2008*). Olfaction is key to the survival of *Drosophila* larvae: attractive odors guide larvae towards food sources (*Asahina et al., 2008*; *Gomez-Marin and Louis, 2012*). Feeding in larvae is also facilitated by the aggregation of groups of larvae that expel digestive enzymes in their saliva — a process called social digestion (*Gregg et al., 1990*; *Sakaguchi and Suzuki, 2013*). Spatial aggregation of larvae relies on the detection of pheromones (*Mast et al., 2014*; *Del Pino et al., 2015*). Our results demonstrate that *Drosophila* larvae can exploit aqueous olfactory cues (*Figure 5*). For both *D. melanogaster* and *D. suzukii*, the gradient of an attractive odor, ethyl acetate, promoted diving at the onset of the trial. In contrast with the sustained attraction elicited by ethyl acetate in gaseous phase (*Figure 7—figure supplement 1*), this odor did not promote sustained diving in *D. suzukii* after a first dive (*Figure 7C*). Considering the fact that *D. suzukii* dives longer (*Figure 6C*) but less frequently than *D. melanogaster* (1.9 ± 1.4 versus 4.1 ± 3.1 in absence of odor), we speculate that *D. suzukii* limits its exploratory dives after having experienced a first search that did not result in the successful discovery of food. By contrast, *D. melanogaster* does not appear to be deterred by unsuccessful dives (*Figure 5C and D*, and *Figure 5—figure supplement 2*). Overall, *D. melanogaster* and *D. suzukii* appear to have evolved different ways to weigh the factors that control the balance between surfacing, digging (exploitation) and exploratory dives. In both species, effective food search necessitates combining chemosensory cues with contextual information related to the physiological state and the history of the animal. Although the neural circuitry that computes the cost-benefit balance remains to be identified, recent work has highlighted the crucial role played by olfaction and nutrient homeostasis in the control of food-search behavior in adult flies (*Corrales-Carvajal et al., 2016*), and the involvement of the mushroom bodies and the lateral horn (*Wang et al., 2013*; *Bräcker et al., 2013*; *Lewis et al., 2015*).

Using the dig-and-dive assay, we examined whether aversive odors could be used as a means to deter larvae from digging and diving into fruits. We tested the effect of geosmin, an odorant volatile associated with the presence of toxic compounds in overripe fruits (*Stensmyr et al., 2012*). In both *D. melanogaster* and *D. suzukii*, geosmin increased their surfacing time. In the case of *D. suzukii*, this trend co-occurred with a significant reduction in diving and active attempts to escape the assay chamber. For *D. suzukii*, we additionally tested the effect of the common insect repellent DEET (N, N-Diethyl-meta-toluamide, *Brown and Hebert, 1997*; *Ditzen et al., 2008*; *Katz et al., 2008*) and found that it elicited significant reduction in diving and that it promoted escapes out of the arena (*Figure 8—figure supplement 2*). Together, these observations highlight the potential of pest control strategies in which the use of non-toxic chemicals could refrain *D. suzukii* larvae from digging and diving, which in turn would slow down feeding and development and increase exposure to parasitoid wasps at the surface of fruits (*Cini et al., 2012*). Such strategies could complement existing efforts to lure adult flies into traps filled with attractive volatile chemicals (*Walsh et al., 2011*; *Abraham et al., 2015*). Furthermore, the dig-and-dive assay appears well suited to identify and to characterize the effects of pheromones involved in aggregation behavior (*Gregg et al., 1990*; *Mast et al., 2014*). This knowledge might guide future efforts to devise methods to interfere with

the ability of larvae to aggregate and to destroy crops by feeding through their dig-and-dive behavior.

## Materials and methods

### Fly stocks

Fly stocks were maintained on conventional cornmeal-agar molasses medium at 22°C in a 12 hr dark–light cycle. Canton-S and *Orco* null mutant (*Orco*$^{-/-}$, RRID:BDSC_23130) (*Larsson et al., 2004*) were used as *Drosophila melanogaster* wild-type and anosmic larvae. Larvae of wild type *Drosophila suzukii* (RRID:FlyBase_FBst0204203) were utilized to study differences in exploratory behavior in hydrogel environments between *Drosophila* species.

### Device fabrication

The template of the dig-and-dive fluidic device was designed in Autodesk Inventor (AutoCAD, Autodesk, San Rafael, CA) and converted into a CAM model (ConstruCAM-3D, Geldern, Germany) for the following fabrication of the device master by micro-end-milling. As shown in *Figure 2A* and *Figure 2—figure supplement 1*, the device comprises six assay chambers. A Computer Numerically Controlled (CNC) milling machine was used to directly cut a polymethylmethacrylate (PMMA) substrate to create a master mold (tools down to 500 μm diameter and speeds of 20000 rpm). A polydimethylsiloxane (PDMS) prepolymer mixture (Sylgard 184, 10:1, Dow-Corning, Midland, MI) was then cast over the mold and cured at 80°C during 1 hr. After curing, the PDMS replica was peeled off from the mold, treated with oxygen plasma (50 W, 30 s) to activate the PDMS surface, and manually bonded to a glass slide (treated with oxygen plasma as well) to seal the chip. The inlet of each assay chamber was closed during the experiments with a PDMS cap. In both sides of the inlet, we made thin air channels (200 μm in thickness) to supply oxygen. Each assay chamber is divided into two sub-chambers excluding the inlet, air and agarose chambers, which are connected in series from the top. As indicated in *Figure 2—figure supplement 1*, the air chamber has a width of 5 mm and a length of 2 mm. The agarose chamber below the air chamber has a width of 3 mm and a length of 12 mm. The thickness of the chamber is 1 mm.

### Establishment of chemical gradient

Bromophenol blue (Sigma-Aldrich, St. Louis, MI) was employed to visualize the gradient and to estimate the diffusion of the odor along the length of the dig-and-dive chamber. We injected 1 μL of BPB (1 mM) at the bottom of agarose chamber (11 mm in depth) using a Hamilton syringe and diffusion of BPB molecules in time was monitored. By taking a picture of the assay, we confirmed that a stiff gradient of BPB was formed after 20 min. The gradient remained stable for 30 min (*Figure 5—figure supplement 1*). The same procedure was applied to establish gradients of odors (40 mM ethyl acetate and 0.5 mM geosmin, Sigma-Aldrich) and an insect repellent, DEET (10 mM, Sigma-Aldrich): the chemicals were diluted in distilled water and injected into chambers 20 min before the introduction of larvae.

### Behavioral experiments: dig-and-dive assay

Experiments were conducted during day time with 3$^{rd}$ instar larvae at a room temperature of 22–24°C. As done in previous work (*Gomez-Marin et al., 2011*), larvae were removed from the food medium by pouring 15% sucrose solution and washed twice in distilled water before the start of experiments. Two devices were superimposed on top of each other on the stage in front of a light pad (Slimlite Lightbox, Kaiser Fototechnik, Buchen, Germany), which illuminated the 12 chambers homogeneously. Twelve larvae were washed and introduced in a separate dive chamber filled with agarose. The movements of individual larvae were recorded at a rate of 1 frame/sec for 15 min. For the experiments involving the block of the posterior spiracles with thermoplastic adhesive (TEC-BOND, Power Adhesives Ltd., Basildon, UK), larvae were treated within 6 min prior to their introduction into the dive chamber.

## Data analysis: dig-and-dive assay

Building on an existing tracking algorithm (*Gomez-Marin et al., 2012*), the centroid position of individual larvae was tracked in the horizontal and vertical axis by using custom-developed software written in Matlab (Mathworks, Natick, MA). All Matlab scripts are available on the Github account of the Louis lab: https://github.com/LabLouis/eLife2017_Dig-Dive (*Kim and Louis, 2017*). A copy is archived at https://github.com/elifesciences-publications/eLife2017_Dig-Dive. The reconstructed trajectories were used to calculate the probability density function (PDF) of larval centroid depths, the percentage of total time that an animal spent in a given behavioral mode during the total assay time (15 min). To classify behavioral modes, the experimental PDF of larval centroid depths was smoothened by applying kernel density estimation. The PDF shown in *Figure 2B* is characterized by a long tail. We assumed that the dominant mode corresponded to digging, whereas the tail with large value corresponded to diving. Accordingly, the threshold value for diving mode ($Z_{diving}$ = −3.6 mm) was determined by taking the first local minimum of the estimated PDF below the mean, $Z$ = −2.7 mm. The same methods were applied to define the threshold value for diving mode in *D. suzukii* ($Z_{diving}$ = −3.4 mm). As we could not easily define the point of overlap between the distributions of surfacing and diving modes in the estimated PDF, we arbitrarily set the threshold value for digging mode to the agarose surface ($Z_{digging}$ = 0 mm). This threshold was used for both species. In the experiments of *D. suzukii* with repellents, when the centroid of a larva was above the top of the air chamber, it was defined as the escaping mode. To quantify the percentage of escape, a given larva was labeled as having escaped as soon as its centroid had been observed out of the arena for more than 30 s. The duration of a single dive was defined as the time that a larva spent a continuous sequence of diving mode. Maximum dive depth was defined as the maximum depth reached during a given dive. Upon completion of each experiment, the behavior of individual larvae was visually inspected through a replay of the movies. The minority of larvae that were inactive (less than 10%) were excluded from the analysis.

## Behavioral experiments and data analysis: quantification of mouth hook contractions during dig-and-dive behavior

The dig-and-dive assay was prepared with two types of agarose gels: 0.4% plain and 0.4% mixed with fructose (1.5 M). Six larvae were loaded in each of the dig-and-dive chamber of the assay (*Figure 2A*). The movements of larvae were filmed for 15 min at a higher temporal resolution of 10 Hz to accurately monitor individual patterns of mouth hook contraction (MHC) (*Video 9*). The MHC was quantified in two different behavioral states, digging and diving. In digging mode, the MHC of each larva was counted over a continuous 60 s time window comprised between the 5th and 10th min from the recording. In diving mode, the MHC of each larva was counted during individual dives.

## Behavioral experiments and data analysis: survival assay

We prepared 10 microcentrifuge tubes (1.5 mL) containing aqueous gels (1 mL) with two different concentrations of agarose: 0.05% and 0.4%. Larvae ($n$ = 8 ~ 14) were introduced in the middle of the agarose gel of each tube. Two small air holes were made in the lid of each tube. After the introduction of the larvae, the cap of the tube was closed. In the 0.4% agarose gels, larvae were able to freely move up and down the gel

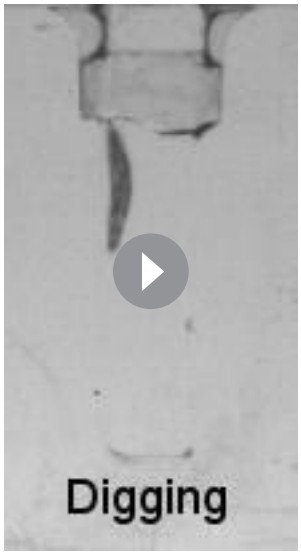

**Video 9.** *D. melanogaster* larva in a dig-and-dive chamber containing 0.4% agarose gel mixed with fructose (1.5 M). A sequence of digging and diving motions is presented (real-time speed). The movie correspond to a sequence of 99 s excerpted from a 15 min trial. Transitions between behavioral modes are indicated by a 2-s banner at the bottom of the movie.
DOI: https://doi.org/10.7554/eLife.27057.032

substrate. Larvae could not move properly in the soft 0.05% gel. As a consequence, the group of larvae introduced in each tube was quickly separated into two subgroups: larvae that had sunk in the middle of the gel and larvae that managed to remain right below the surface of the gel. After 1 hr, larvae were collected from the tubes and transferred to food vials. For the 0.05% gels, the sunk and non-sunk larvae were maintained in separate food vials. The transferred larvae were kept in a 22°C incubator for 10 days before the number of adult flies was counted in the larva. We defined the survival ratio as the number of larvae that developed into adult flies divided by the total number of larvae. For 0.05% and 0.4% agarose gels, a survival ratio was calculated from the total number of larvae found in the vial. In addition, we calculated a survival ratio based on the number of larvae that had sunk in the 0.05% agarose gels.

## Behavioral experiments and data analysis: two-choice preference assay

Petri dishes (90 mm in diameter) with an imprinted quadrant grid were used to prepare assay plates. Warm liquefied agarose solution of one type was poured into the dish to form a layer of 5 mm in height. After solidification, the first and third quadrants were removed by a razor blade. Then, warm liquefied agarose solution of a second type was poured into the empty quadrants up to a height equal to the other two existing quadrants. The types of agarose solutions used throughout the paper were as follows: 0.5% plain, 1.0% plain, 2.0% plain, 0.5% and 2.0% mixed with the mesocarp-tomato juice (50% v/v), 2.0% mixed with tomato juice obtained from both the locular gel and mesocarp (50% v/v), 2.0% mixed with fructose (1.5 M). Gels prepared with agarose concentrations of 0.5% and 2.0% have the similar ranges of mechanical hardness as the locular gel and the mesocarp of tomatoes, respectively (*Grant et al., 2012*; *Li et al., 2012*). To prepare the two different types of tomato juices tested in the study, the locular gel and the mesocarp of regular tomatoes were harvested and ground in a blender. At the onset of the experiment, a group of larvae ($n$ = 20 for *D. melanogaster* and $n$ = 10 for *D. suzukii*) were placed in the middle of the assay plate. After 3 min, the number of larvae on each quadrant was counted. The preference of larvae for one of the two quadrant types was quantified through the following preference index: PI = (total number of larvae on the condition A – total number of larvae on the condition B)/total number of all larvae.

## Statistical procedures

The normality of each sample was assessed by using the Lilliefors test. When the test rejected the normality of a given sample with a confidence level higher than 95%, nonparametric tests were applied to compare the medians of this sample with other samples (Mann-Whitney test for pairwise comparisons and Kruskal-Wallis for comparisons of multiple samples). When the normality of samples involved in an experiment was confirmed, a two-tailed Student *t*-test was applied for pairwise comparisons. Whenever one of the samples to be compared was not normally distributed, we applied nonparametric statistics to treat the results of the entire collection of samples of this experiment.

For samples distributed according to normal distributions, the 'sampsizepwr' function from the Statistics toolbox in Matlab (Mathworks, Natick, MA) was used for sample-size estimations. To estimate the sample size ($n_0$), we combined a standard power level of 0.8 with a suspected mean ($\mu$) and standard deviation ($\sigma$), and desired difference ($\delta$) (*Zar, 1999*). Suspected means, standard deviations and $\delta$ differences were derived from pilot experiments conducted with the dig-and-dive assay. To be on the conservative side, the final sample size for each experiment was chosen to be larger than the sample-size estimation. Through this procedure, we inferred that the sample size ($n_0$) of the tests involving the dig-and-dive assay should be larger than 20. Since each assay contains six individual chambers (trials), we decided to conduct four repetitions to obtain a sample size of 24 independent trials for each experiment.

For samples distributed normally, the power of each statistical test was calculated with the sample size, mean, and standard deviation of the control group, and the difference between the control and test groups. For the experiments with more than one pairwise comparison, the power analysis was based on the least favorable pairwise comparison. We ensured that sample sizes were sufficiently large so that the power of each test exceeded 0.8. Throughout the study, we used Cohen's *d* (absolute difference between means divided by the standard deviation) to control for the effect size for tests on differences in means (*Sheskin, 2003*). We observed a large difference effect (Cohen's

d > 0.5) for all *t*-tests leading to a rejection of the null hypothesis, except one test in *Figure 5—figure supplement 5*, which was associated with a moderate effect (Cohen's d = 0.48). Power analysis could not be achieved for comparisons involving samples that departed from normal distributions (*Sheskin, 2003*).

For the behavioral experiment in a tomato slice (Student's *t*-test, *Figure 1A'*), we calculated a sample size $n_0$ of 6 using the following set of parameters inferred from pilot experiments: $\mu = 65$, $\sigma = 10$, and $\delta = 10$. The final sample size was brought to eight repetitions. The power of the test at time point t = 2 min was found to be 1, validating the appropriateness of the final sample size. For the survival assay of *Figure 3—figure supplement 1B*, we obtained a sample size of $n_0 = 6$ using Student's *t*-test with $\mu = 0.5$, $\sigma = 0.2$, and $\delta = 0.3$. The final size was brought to 10. The power of this test was found to be 1, validating the appropriateness of the final sample size. For *Figure 7D*, we obtained a sample size of $n_0 = 34$ using Student's *t*-test with $\mu = 0.15$, $\sigma = 0.1$, and $\delta = 0.05$. With a final sample size of 48, the power of this test was found to be 1. For the quantification of the percentages of time spent in a particular behavioral mode in the dig-and-dive assay, samples significantly departed from normal distributions and the previous power analysis did not apply. If the samples of the diving times of *D. melanogaster* in 0.2% and 0.4% agarose chambers (*Figure 3C*) were assumed to be normally distributed, a *t*-test would be associated with a sample size $n_0 = 20$ for parameters $\mu = 4$, $\sigma = 15$, and $\delta = 11$. The power of this test would be 1, thereby suggesting the appropriateness of a sample size of 24 for non-parametric statistics. The same logic was applied to justify the sample size of *Figure 7—figure supplement 1*: $n_0 = 12$ was found to be appropriate for a Student's *t*-test with $\mu = 5$, $\sigma = 15$, and $\delta = 15$. The final sample size was brought to 15. The power of the corresponding *t*-test was 1, indirectly confirming the appropriateness of the final sample size. The sample size and p-values of the individual tests carried out in the study are reported in the figure captions and *Supplementary file 1*.

## Acknowledgements

We are deeply thankful to B Iyengar for conceiving and testing a pilot version of the dig-and-dive assay. We are grateful to A Gomez-Marin for help in developing the tracking software, S Kraus for having experimentally tested the pilot version of the assay, and A Schulze for help in estimating the stability and the geometry of the odor gradient, Y-J Kim and his lab members for their kind hospitality and support. We thank C Mirth, J Simpson, M Stensmyr, A Gomez-Marin and the Louis lab for comments on the manuscript. We acknowledge support of the Spanish Ministry of Economy and Competitiveness, 'Centro de Excelencia Severo Ochoa 2013–2017', as well as the support of the CERCA Programme/Generalitat de Catalunya. This work was supported by funding from the Spanish Ministry of Science and Innovation (MICINN, BFU2008-00362 and BFU2011-26208), the EMBL/CRG Systems Biology Program and the EU FET-Open grant MINIMAL.

## Additional information

### Funding

| Funder | Grant reference number | Author |
| --- | --- | --- |
| European Commission | EU FET-OPEN MINIMAL | Daeyeon Kim<br>Matthieu Louis |
| Generalitat de Catalunya | Cerca | Daeyeon Kim<br>Mar Alvarez<br>Laura M Lechuga<br>Matthieu Louis |
| Ministerio de Economía y Competitividad | BFU2008-00362 | Daeyeon Kim<br>Mar Alvarez<br>Laura M Lechuga<br>Matthieu Louis |
| Ministerio de Economía y Competitividad | BFU2011-26208 | Daeyeon Kim<br>Mar Alvarez<br>Laura M Lechuga<br>Matthieu Louis |

The funders had no role in study design, data collection and interpretation, or the decision to submit the work for publication.

## Author contributions

Daeyeon Kim, Conceptualization, Software, Formal analysis, Investigation, Visualization, Methodology, Writing—original draft; Mar Alvarez, Resources, Methodology; Laura M Lechuga, Supervision, Funding acquisition, Methodology; Matthieu Louis, Conceptualization, Supervision, Funding acquisition, Investigation, Methodology, Writing—original draft

## Author ORCIDs

Mar Alvarez (iD) http://orcid.org/0000-0003-4590-4401
Matthieu Louis (iD) http://orcid.org/0000-0002-2267-0262

## Decision letter and Author response

Decision letter https://doi.org/10.7554/eLife.27057.036
Author response https://doi.org/10.7554/eLife.27057.037

# Additional files

## Supplementary files

• Supplementary file 1. Tables summarizing the conditions and results of the statistical tests conducted throughout the manuscript.
DOI: https://doi.org/10.7554/eLife.27057.033

• Transparent reporting form
DOI: https://doi.org/10.7554/eLife.27057.034

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
