## [Decision Letter]

Thank you for submitting your article "Species-specific modulation of food-search behavior by respiration and chemosensation in *Drosophila* larvae" for consideration by *eLife*. Your article has been reviewed by three peer reviewers, and the evaluation has been overseen by a Reviewing Editor and K VijayRaghavan as the Senior Editor. The reviewers have opted to remain anonymous.

The reviewers have discussed the reviews with one another and the Reviewing Editor has drafted this decision to help you prepare a revised submission.

Summary:

The manuscript by Kim et al. entitled "Species-specific modulation of food-search behavior by respiration and chemosensation in *Drosophila* larvae" uses *Drosophila* larvae as a model system to analyze the food-search behavior of animals. The authors have established a new behavioral assay system that they call the "dig-and-dive" paradigm, which is useful in studying larval exploration behavior along the vertical axis. Three behavioral modes (surfacing, digging, and diving) were defined and quantified in this assay. The authors show that the hardness of the substrate, oxygenation level, and odors (attractive and repulsive) influence larval food-search behavior. In particular, by quantifying the behavior of two Drosphila species, *D. melanogaster* and *D. suzukii*, the authors explain the behavioral differences between these two species under natural conditions.

Innovative assays that better mimic natural ecologically relevant conditions, without compromising quantifiability, are important steps to overcome current limitations in laboratory behavioral studies. Kim et al. provide a very exciting example of such an effort. The manuscript is clearly written and a pleasure to read. The behaviors are described in meticulous details and with quantitative rigor.

As the authors show experimentally, detection of texture (hardness) is an important factor that determines dig-and-dive behavior. This leads to the question of how mutants of genes involved in mechanosensation behave in response to substrates of different hardnesses, since it is possible that the increase of *D. suzukii* digging behavior in hard food (2% agarose) could be related to genetic variation in mechanosensation. However, as the authors discuss, such analyses are probably the topic of future studies, and this paper constitutes a complete story by itself, in establishing an assay system useful for such studies.

Essential revisions:

1) For the experiments with tomatoes: is the soft part more nutritious than the harder part? To what extent is the attraction to the soft part due to the differences in the nutritional value as opposed to textural differences like hardness? It is known that big cats first eat the soft inner organs of their prey, which presumably has greater nutritional value as well.

2) Do *Drosophila suzukii* still prefer hard surfaces to lay their eggs if given a choice with a softer surface fruit? For example, between fresh and rotting cherries? What about larval behavior: do *suzukii* prefer harder surface when given a choice between soft and hard substrate?

3) Mouth hook movements do not necessarily correlate with food intake, since these are also used for crawling and locomotion. How strict is the correlation between mouth hook movements and actual food intake in their assay? Do larvae change their mouth hook movements and decouple it with food intake when faced with increased drowning probability?

---

## [Author Response]

Summary:

The manuscript by Kim et al. entitled "Species-specific modulation of food-search behavior by respiration and chemosensation in Drosophila larvae" uses Drosophila larvae as a model system to analyze the food-search behavior of animals. The authors have established a new behavioral assay system that they call the "dig-and-dive" paradigm, which is useful in studying larval exploration behavior along the vertical axis. Three behavioral modes (surfacing, digging, and diving) were defined and quantified in this assay. The authors show that the hardness of the substrate, oxygenation level, and odors (attractive and repulsive) influence larval food-search behavior. In particular, by quantifying the behavior of two Drosphila species, D. melanogaster and D. suzukii, the authors explain the behavioral differences between these two species under natural conditions.

Innovative assays that better mimic natural ecologically relevant conditions, without compromising quantifiability, are important steps to overcome current limitations in laboratory behavioral studies. Kim et al. provide a very exciting example of such an effort. The manuscript is clearly written and a pleasure to read. The behaviors are described in meticulous details and with quantitative rigor.

As the authors show experimentally, detection of texture (hardness) is an important factor that determines dig-and-dive behavior. This leads to the question of how mutants of genes involved in mechanosensation behave in response to substrates of different hardnesses, since it is possible that the increase of D. suzukii digging behavior in hard food (2% agarose) could be related to genetic variation in mechanosensation. However, as the authors discuss, such analyses are probably the topic of future studies, and this paper constitutes a complete story by itself, in establishing an assay system useful for such studies.

We agree that studying the factors that condition the ability of *D. suzukii* to dig and dive into harder substrates than *D. melanogaster* represents a fascinating follow-up of the present study. Nailing down the relevant morphological and locomotor differences between both species will clarify how evolution acts to produce behavioral specialization. Mechanosensation and the neuromuscular systems are excellent candidates to display evolutionary differences. However a thorough comparative study would require work that goes beyond the scope of the present manuscript. Such a comprehensive study was recently achieved for egg-laying in *D. suzukii* (Karageorgi et al., 2017).

Essential revisions:

1) For the experiments with tomatoes: is the soft part more nutritious than the harder part? To what extent is the attraction to the soft part due to the differences in the nutritional value as opposed to textural differences like hardness? It is known that big cats first eat the soft inner organs of their prey, which presumably has greater nutritional value as well.

We have conducted a series of new experiments to examine whether *D. melanogaster* larvae show a preference for the locular-gel layer of tomatoes due to its stiffness or intrinsic nutritive properties. We introduced a preference assay in which a group of larvae is given a choice between two gels of equal stiffness mixed with extracts from either the locular or the mesocarp layers of regular tomatoes. Larval choice was studied in Petri dishes subdivided into 4 quadrants. The type of gel comprised in each quadrant could be laid independently. When larvae were tested for their responses to the locular and the mescocarp extracts in the absence of any differences in the substrate’s stiffness, they demonstrated no preference to any for two extracts. This result indicates that possible differences in the nutritive contents of the locular gel and mescocarp are insufficient to explain the preference for the locular gel. As reported in the following references (Grant et al., 2012; Li et al., 2012), the locular layers correspond to a hydrogel with a concentration equivalent to ~0.5%. By contrast the mesocarp is much harder and it corresponds to an agarose gel at a concentration of ~2.0%. Based on this information, we tested the preference of larvae for mesocarp extracts mixed in a gel of either 0.5% agarose (soft) or 2.0% agarose (hard). Larvae demonstrated a marked preference for the soft gel, which is sensible given that digging into softer substrates must entail a lower metabolic cost. These results are now presented in the new Figure 1—figure supplement 1. We conclude the preference of *D. melanogaster* larvae for the locular gel of tomatoes is due to the textural properties of this layer.

2) Do Drosophila suzukii still prefer hard surfaces to lay their eggs if given a choice with a softer surface fruit? For example, between fresh and rotting cherries? What about larval behavior: do suzukii prefer harder surface when given a choice between soft and hard substrate?

Recently, Prud’homme, Gompel and coworkers published an elegant study that documents the ability of *D. suzukii* to lay eggs in substrates much stiffer than *D. melanogaster*. This aptitude explains one ecological advantage of *D. suzukii* over other species: by laying eggs in fresh fruits, *D. suzukii* ensures that its progeny populates food resources before they become suitable egg-laying sites for other *Drosophila* species. Despite the ability of *D. suzukii* to lay eggs in harder substrates than *D. melanogaster, D. suzukii* was found to prefer feeding on softer than harder fruits (Karageorgi et al., 2017). In the new Figure 6—figure supplement 1, we tested the preference of *D. suzukii* larvae for soft (0.5% agarose) versus hard (2.0% agarose) substrates in a two-choice preference assay set in a Petri dish. Like their adult counterparts, *D. suzukii* larvae displayed a clear preference for soft substrates. Therefore the ability of *D. suzukii* larvae to cope with harder substrates does not appear to have co-evolved with a shift in preference for hard substrates in choice conditions.

3) Mouth hook movements do not necessarily correlate with food intake, since these are also used for crawling and locomotion. How strict is the correlation between mouth hook movements and actual food intake in their assay? Do larvae change their mouth hook movements and decouple it with food intake when faced with increased drowning probability?

In new experiments, we explored the correlation between feeding behavior and the dig-and-dive behavior. This question could not be addressed in very soft substrates where the probability of drowning is high since larvae refrained from digging and diving under such conditions and they spent most of their time surfacing (see 0.05% substrate hardness in Figure 3). In the dig-and-dive assay, monitoring the mouth of hook contractions (MHC) of larvae on the surface is technically difficult and inaccurate. As an alternative, we monitored the behavior of immersed larvae exposed to food mixed with the substrate. The presence of food (sugar) has been shown to promote food ingestion through MHC, as documented by Shen and colleagues (Wang, Pu and Shen, 2013). Although we could not assess the instantaneous rate of food intake in the dig-and-dive assay, differences in food intake could be indirectly inferred from relative changes in the rates of MHC observed in presence and in absence of food. Following this logic, the rate of MHC was quantified manually from new high-resolution movies in both behavioral modes (see illustrative movie sequence in Video 9).

In Figure 5—figure supplement 3, we had already established that yeast — a natural food source for larvae — promotes a shift from diving to digging. Unfortunately yeast-containing gels were slightly opaque, which precluded accurate tracking of the behavior of the larva. For this reason, we tested the behavior of larvae in agarose gels mixed with sugar (fructose), which produces strong attraction (see new Figure 5—figure supplement 4) and is sufficient to act as an appetitive reward in classical conditioning experiments (Gerber and Stocker 2007). When placed in a sugar-containing substrate, the frequency of diving significantly decreased like in the presence of yeast (see new Figure 5—figure supplement 4). The rates of MHC in the diving mode did not differ significantly in gels with and without sugar (see new Figure 5—figure supplement 5), suggesting nearly no feeding activity. In agreement with the reviewer’s assertion, this basal rate of MHC might be part of the locomotor program that permits larvae to propel themselves through the agarose gel during dives. By contrast, the rate of MHC observed in digging mode significantly increased in presence of sugar. Larvae did not actively displace their whole body in digging mode, implying that the MHC might be mainly used for feeding. So, the change in MHC is likely to be due to an increase in the ingestion of food triggered by the detection of sugar. In conclusion, our observations indicate that digging is a behavioral mode that enables larvae to feed in conditions safer than on the surface. MHC during digging mostly serves the purpose of feeding without generating locomotion. Diving is a form of exploratory behavior that appears to be independent of active feeding. In line with the reviewer’s suggestion, MHC can be uncoupled from food intake during diving where it contributes to the locomotor program.